# Thermal Analysis of Aliphatic Polyester Blends with Natural Antioxidants

**DOI:** 10.3390/polym12010074

**Published:** 2020-01-02

**Authors:** Olga Olejnik, Anna Masek, Adam Kiersnowski

**Affiliations:** 1Institute of Polymer and Dye Technology, Faculty of Chemistry, Lodz University of Technology, ul. Stefanowskiego 12/16, 90-924 Lodz, Poland; 183176@edu.p.lodz.pl; 2Das Leibniz-Institut für Polymerforschung Dresden e. V. (IPF), Hohe Str. 6, D-01069 Dresden, Germany; adam.kiersnowski@pwr.edu.pl; 3Faculty of Chemistry, Wroclaw University of Science and Technology, Wybrzeze Wyspianskiego 27, 50-370 Wroclaw, Poland

**Keywords:** polymer blends, natural stabilizers, phytochemicals

## Abstract

The aim of this research was to enhance thermal stability of aliphatic polyester blends via incorporation of selected natural antioxidants of plant origin. Thermal methods of analysis, including differential scanning calorimetry (DSC) and thermogravimetry (TGA), are significant tools for estimating the stabilization effect of polyphenols in a polymer matrix. Thermal stability was determined by analyzing thermogravimetric curves. Polymers with selected antioxidants degraded more slowly with rising temperature in comparison to reference samples without additives. This property was also confirmed by results obtained from differential scanning calorimetry (DSC), where the difference between the oxidation temperatures of pure material and polymer with natural stabilizers was observed. According to the results, the materials with selected antioxidants, including *trans*-chalcone, flavone and lignin have higher oxidation temperature than the pure ones, which confirms that chosen phytochemicals protect polymers from oxidation. Moreover, based on the colour change results or FT-IR spectra analysis, some of the selected antioxidants, including lignin and *trans*-chalcone, can be utilized as colorants or aging indicators. Taking into account the data obtained, naturally occurring antioxidants, including polyphenols, can be applied as versatile pro-ecological additives for biodegradable and bio-based aliphatic polyesters to obtain fully environmentally friendly materials dedicated for packaging industry.

## 1. Introduction

Nowadays, many scientists are focused on creating non-toxic, bio-based and biodegradable materials [1,2,3,4]. In such composites, additives play an important role and should also have positive impact on natural environmental and human health. The application of selected phytochemicals as additions to polymers with biodegradability and derived from renewable resources is an innovative way to obtain pro-ecological polymeric materials [5,6,7,8].

Phytochemicals are substances with a variety of physical, chemical and biological characteristics that play important roles in plant growth and development [9,10,11]. The majority of them have antioxidant properties and their role relies on protecting against the harmful impact of free radicals. These substances, called antioxidants, are able to delay or inhibit oxidation reactions which are responsible for the aging processes [11]. Many of the phytochemicals are, because of their beneficial properties, used as food supplements, bioactive pharmaceutical components and additives for cosmetics and perfumes [12,13]. A diet enriched with vegetables and fruit, which are full of natural antioxidants, has beneficial impact on human health, delaying the organism’s ageing processes, decreasing inflammation and reducing risk of cancer developing [14,15].

The many advantages of phytochemicals also led to using them in polymer technology. Antioxidants derived from plants act similarly to stabilizers commonly used for different materials. The main role of these compounds is to protect from oxygen action and its reactive forms. Active phytochemicals prevent free radical (oxidant) creation. Natural stabilizers are able to inhibit the initiation step of the metal oxidation process, including cadmium, mercury, copper and lead. Moreover, these compounds are able to intercept created oxidants and inhibit chain reactions which cause the formation of more radicals [5,16].

The most common antioxidants derived from plants are carotenoids, such as carotenes or xanthophylls. Also, natural phenolic compounds, including phenolic acids, polyphenols and their polymeric forms (tannins, lignins) belong to the effective antioxidants of plant origin [5,17]. Natural compounds with antioxidant properties act with different mechanisms, counteracting the oxidation processes. In case of carotenoids, the quenching of singlet oxygen [5] and electron transfer, as well as radical (peroxide and thiol) adduct creation, take place [18]. On the other hand, phenolic compounds are able to transfer hydrogen atoms and chelate transition metals [5,19].

Some of the antioxidants are capable, not only of counteracting the reactive forms of oxygen, but also absorbing UV radiation. Thus, these compounds prevent photodegradation and belong to the more effective stabilizers [20,21]. Quercetin can be presented as one of such UV absorbers with antioxidant properties. This compound belongs to the flavonoid group of antioxidants, more precisely flavonols [22]. Capacity of this substance to counter UV rays has been proven in polypropylene (PP) [23], as well as in biodegradable polymers, including polylactide (PLA) matrix [24]. Other flavonoids, like chrysin, hesperidin and naringenin can also be used as photo-stabilizers [23].

Many antioxidants can also play a role as thermal stabilizers. One of the natural thermal stabilizers with antioxidant properties is quercetin [23]. Another antioxidant with similar characteristics is 2-hydroxy-1,4-naphthoquinone, also called lawsone, which is a component of the popular natural colorant, henna. It has been proven that vulcanizates of ethylene-propylene elastomer that contain this substance show higher thermal stability than reference samples. Furthermore, lawsone turned out to also be a good UV absorber [25].

Synergistic or antagonistic stabilization effect can be obtained by incorporating more than one antioxidant. An interesting idea was to utilize waste materials derived from agriculture, which are rich in natural antioxidants. Selected agro-waste, including grape pomace waste, turmeric shavings and waste, coffee grounds, and orange peel waste proved to be good stabilizers to protect polyolefins from thermo-oxidative aging. Furthermore, no chemical treatment or extraction was needed [26]. A similar effect was observed with waste products from tomato and wine processing incorporated into polypropylene matrix [27]. Essences that contain mixtures of various antioxidants, including green or black tea extracts, turned out to be good stabilizers for polypropylene [28]. On the other hand, coffee, cocoa and cinnamon extracts effectively stabilize aliphatic polyesters [8]. Also, mixtures of polyphenols derived from *Cistus linnaeus* and *Juglans regia Linnaeus* walnut green husk incorporated into a bio-based and biodegradable polymeric matrix via an impregnation process work as environmentally friendly stabilizers [29].

The examples presented concerning beneficial impact of natural antioxidants on polymers indicate that there is a real possibility to apply these substances as pro-ecological additives for polymeric materials. This is a relatively innovative and new area of research which should be continued and developed. Many of the antioxidants derived from plants have, not only stabilizing properties, but also are characterized by a variety of other features, such as colouring ability. The application of natural stabilizers as additives for polymers, including bio-based and biodegradable materials, leads to creation of totally environmentally friendly composites which is highly demanded nowadays. This research contributes to extending the knowledge about this innovative topic and develop this branch of science. According to the present studies, other different natural antioxidants, including flavone, *trans*-chalcone and lignin can become pro-ecological additives for polymers.

## 2. Materials and Methods

### 2.1. Reagents

The object of the research was to study polymer blends based on polylactide (PLA) and polyhydroxybutyrate (P(3.4 HB) contained selected natural antioxidants. Polylactide (PLA), Ingeo 4043D, was obtained from NatureWorks LLC (Minnetonka, MN, USA). This material with a density of 1.25 g/cm^3^ is characterized by glass transition of about 55–60 °C. The melting point of this polymer is 145–160 °C and the melt flow index equals 6 g/10 min. Poly(hydroxybutyrate) (P(3.4 HB)) containing 12 mol% 4-hydroxybutyrate was obtained from Simag Holdings LTD (Hong Kong, China). The density of this polymer is 1.25 g/cm^3^, its melt flow index equals 18 g/10 min and the melting point is 170 °C. Different natural antioxidants, including lignin (alkali, low sulfonate content lignin, kraft, pH: 10.5 (3 wt %)), flavone (*M*_w_ = 222.24 g/mol, mp: 94–97 °C, ≥99.0%) and *trans*-chalcone (*M*_w_ = 208.26 g/mol, mp: 55–57 °C, 97%), used as stabilizers, were obtained from Sigma-Aldrich (Saint Louis, MI, USA).

### 2.2. Sample Preparation

In the first stage, samples were prepared by melt blending in a DSM-5 twin-screw compounder (Xplore^®^, Sittard, The Netherlands) with mixing time of 2.5 min and speed of 100 rpm. The mixing temperature was 170 °C in the case of polyhydroxybutyrate (P(3.4 HB)) reference material and 190 °C for PLA reference material and (PLA/P(3.4 HB)) blends with selected antioxidants. Samples of about 8–9 g were then hot pressed at 170 °C (P(3.4 HB)) referential sample) and 190 °C (PLA and PLA/P(3.4 HB) specimens) with pressing time of 2.5 min. The thin square plates obtained had dimensions of 80 × 80 × 1 mm. The compositions of prepared blends are presented in Table 1.

### 2.3. Differential Scanning Calorimetry (DSC)

Differential scanning calorimetry (DSC) was used for determining the temperature ranges of the characteristic phase changes, glass transition temperature (*T*_g_), cold crystallization temperature (*T*_cc_) and melting temperature (*T*_m_), as well as oxidation temperature (*T*_o_). Enthalpy changes (Δ*H*) were also analysed. The tests were carried out using a Mettler Toledo^®^ DSC1 instrument (TA 2920; TA Instruments, Greifensee, Switzerland) equipped with a STARe System and Gas Controller GC10. The analyser was calibrated on the basis of n-octane and indium standards. The samples of about 8–9 mg, placed in open, aluminum crucibles, were heated from 0 °C to 200 °C at a rate of 20 °C/min under an argon atmosphere. After 10 min at 200 °C, the samples were cooled to 0 °C and the gas was switched from argon to air at a flow rate of 50 mL/min. Finally, samples were heated to 350 °C and the curves obtained were analysed.

### 2.4. Themogravimetric Analysis

Thermogravimetric Analysis (TGA) was utilized for detecting the thermal degradation process which is related to the mass loss of the specimen as a function of rising temperature. The analysis was performed by means of a TGA/DSC1 device provided by Mettler Toledo^®^ (TA Instruments, Greifensee, Switzerand), with prior calibration using indium and zinc as standards. The test was carried out in a temperature range of 25–600 °C with heating rate of 15 °C/min under an argon atmosphere at a flow rate of 45 mL/min. The crucibles used in this analysis with volume of 70 μL were made of ceramic (polycrystal aluminium oxide).

### 2.5. Dynamic Mechanical Analysis

The rheological measurements in the solid state were carried out by means of an ARES G2 rotational rheometer (TA Instruments, New Castle, DE, USA) using a small amplitude oscillatory frequency and a temperature ramp within the range of −100 until 150 °C. The heating rate was set at 5 K/min. A frequency of 1 Hz was used and the strain amplitude was 0.05%. All measurements were carried out using nitrogen as the heating gas. Both storage and loss modulus, were measured as a function of the temperature respectively.

### 2.6. Solar Aging

Samples were aging using Atlas SC340 MHG Solar Simulator climate chamber (AMETEK, Inc., Berwyn, IL, USA) with a 2500 W MHG lamp. The solar radiation intensity is reported to be 1200 W/m^2^ at 100% lamp power intensity. Accelerated ageing of the samples was carried out in an Atlas SC340 MHG Solar Simulator climate chamber with a 2500 W MHG lamp. The solar radiation intensity is reported to be 1200 W/m^2^ at 100% lamp power intensity. The specimens were tilted a bit from the horizontal (8° ± 2°) in order to let the water run off the surface. During the ageing period the UVA and UVB radiation intensity was measured at various times to be lying within the interval 60–80 W/m^2^ (a 6% UVA fraction (like in sunlight) of the total solar intensity (1200 W/m^2^) yields 72 W/m^2^ UVA radiation) and 3–6 W/m^2^, respectively. The UV measurements were performed with a radiometer/photometer with an UVA sensor and an UVB sensor. The exposure duration consisted of 30 whole cycles of 24 h, each cycle divided into 20 h with a solar radiation intensity of 1200 W/m^2^ and 4 h with no solar radiation exposure. The solar aging lasted 265 h, with the temperature of *T* = 60 °C and the humidity of 70%.

### 2.7. Mechanical Tests

Static mechanical properties, tensile strength (*TS*) and elongation at break (*E*_b_), were determined with a Zwick/Roell Z2.5 test machine provided by ZwickRoell GmbH & Co. KG (Ulm, Germany) before and after solar aging. Tests were carried out on “dumbbell” shaped samples type 5B, approximately 1 mm thick and 1.7 mm wide centre portion, according to DIN EN ISO 527-2/5B/5. The samples were cut from prepared plates using a stamp. The speed of test was 5 mm/min. The pre-stress was about 0.05 N. The number of individual samples used in this research was 6 for every tested material. Basing on the obtained results, the aging coefficient (A*_f_*) was calculated from the presented equation [30]:(1)$$A_{f}={\left(TS\cdot E_{B}\right)}_{after\;aging}/{\left(TS\cdot E_{B}\right)}_{after\;aging}$$
where *A_f_* is the aging coefficient, *TS* is tensile strength (MPa), and *E_B_* is elongation at break (%).

### 2.8. Fourier Transform Infrared Spectroscopy (FT-IR) Absorbance Spectra Analysis

Fourier transform infrared spectroscopy (FT-IR) absorbance spectra was obtained in the range of 4000–400 cm^−1^ taking advantage of Thermo Scientific Nicolet 6700 FT-IR spectrometer with diamond Smart Orbit ATR sampling equipment. The number of used scans amounted to 64 at a resolution of 4 cm^−1^. In this research the structural changes of tested materials which occurred as a result of ageing processes were investigated by the difference of the peak intensity at 1748 cm^−1^ and the reference peak at 1451 cm^−1^ [31]. The comparison of –C–O (1187cm^−1^) and –C=O (1748 cm^−1^) absorbance was also determined [31,32].

### 2.9. Colour Measurement

The optical investigation of samples was performed before and after solar aging by means of colour measurement. The colour of PLA/P(3.4 HB) blends containing selected antioxidants was measured on the basis of PN-EN ISO 105-J01 using a Spectrophotometer UV-VIS CM-36001 (Konica Minolta Sensing, Inc., Osaka, Japan). This device measures the signal reflected from the surface of the sample and converts it into the impression of colour that is perceived by the human eye. The results were described with the CIE-Lab system (L: lightness, *a*: red-green, *b*: yellow-blue). Moreover, colour difference (Δ*E*), whiteness index (*W_i_*), chroma (*C_ab_*) and hue angle (*h_ab_*) values were calculated according to Equations (2)–(5). The values Δ*a*, Δ*b* and Δ*L* used in these equations were calculated as the difference of *a*, *b* and *L* parameters between samples with and without natural stabilizer. These parameters are useful for estimating polymer colour change after incorporating natural antioxidants to polyester blend matrix [33].
(2)$$∆E=\sqrt{{\left(\Delta a\right)}^{2}+{\left(\Delta b\right)}^{2}+{\left(\Delta L\right)}^{2}}$$
(3)$$W_{i}=100-\sqrt{a^{2}+b^{2}+{\left(100-L\right)}^{2}}$$
(4)$$C_{ab}=\sqrt{a^{2}+b^{2}}$$
(5)$$h_{ab}\{\begin{array}{c} arctg\left(\frac{b}{a}\right),\;when\;a>0\;\cap b>0\\ 180{}^{\circ}+arctg\left(\frac{b}{a}\right),\;when\;\left(a<0\cap b>0\right)\cup (a<0\cap b<0)\\ 360{}^{\circ}+arctg\left(\frac{b}{a}\right),\;when\;a>0\cap b<0 \end{array}$$

## 3. Results and Discussion

### 3.1. Differential Scanning Calorimetry (DSC)

Differential Scanning Calorimetry was useful for oxidation stability detection of prepared materials. Moreover, some thermal parameters, such as glass transition temperature (*T*_g_), cold crystallization temperature (*T*_cc_) and melting point (*T*_m_), were determined. As can be seen in the curve (Figure 1), the blend based on polylactide (PLA) and polyhydroxybutyrate (P(3.4 HB)) has one glass transition temperature, which is higher than for P(3.4 HB) material and lower than for polylactide (PLA). The glass transition temperature (*T*_g_) of prepared blends was about 54 °C and this result corresponds with the data obtained by Zhang et. al. [34]. Adding antioxidants, including flavone and *trans*-chalcone, did not change this parameter significantly. Only lignin turned out to be a type of plasticizer and caused a decrease of *T*_g_ to 52 °C. Based on the results obtained by Gordobil et. al., this phenomenon was not observed with pure PLA, where the addition of different types of lignin did not change this parameter [35]. Moreover, the phenomenon of cold crystallization, which is characteristic of materials crystallizing very slowly, such as PLA or P(3.4 HB), has been detected. Prepared blends of PLA and P(3.4 HB) crystallize quicker and have the highest enthalpy of this process, amounting to 24 J/g, which is presented in Table 2. This is much more than for the referential samples of PLA or P(3.4 HB). This enthalpy might be a result of two types of crystal structure creation, where dispersed P(3.4 HB) crystals act as nucleating agents in PLA matrix [36]. The crystallization temperatures of PLA/P(3.4 HB) blends are between that of pure PLA and P(3.4 HB), furthermore, the addition of flavonoids did not affect this parameter substantially. In the DSC run, the melting point of PLA/P(3.4 HB) materials can also be detected. For the polylactide/polyhydroxybutyrate blend, a bimodal melting peak is observed. According to the data obtained from DSC measurements, two types of crystal are present in this blend. The first stage of melting is related to one type of crystals melting, which probably belongs to PLA. The second one belongs to the P(3.4 HB) crystals, which also corresponds with data given in literature [34,36]. As can be seen, the addition of polyphenols did not change this parameter significantly but it caused an increase in the temperature of the oxidation process. This parameter is related to improvement of material resistance to oxidation. Thanks to flavone, the temperature of the start of the oxidation process in PLA/P(3.4 HB) blend was raised from 233 °C to 241 °C, hence improving the material’s stability. Adding *trans*-chalcone to PLA/P(3.4 HB) blend caused an increase in the temperature of oxidation from about 232 °C to 269 °C, which also made the blends more thermally stable. A similar phenomenon was observed in the case of other antioxidants, including quercetin [23] and lawsone [25].

### 3.2. Thermogravimetric Analysis (TGA)

Thermogravimetric analysis allowed a determination of material thermal stability. According to the TGA thermogram (Figure 2) and Table 3, blending PLA with P(3.4 HB) reduces the resistance of PLA to high temperatures. The blend lost 5% of its weight at a temperature of 284.5 °C, while the temperature of 5% PLA decomposition amounted to 328.7 °C. Adding polyphenols to PLA/P(3.4 HB) blend caused the improvement in thermal stability At first glance, the stability is not noticeable and PLA/P(3.4 HB) blend with antioxidants loses 5% of its weight at a similar temperature as the pure blend. Nevertheless, the added natural stabilizers subsequently slowed down the blend decomposition. This phenomenon starts to occur in case of 15% material weight loss and is the most visible for 50% of sample weight loss. According to the gathered data, PLA/P(3.4 HB) with *trans*-chalcone is the most thermally stable and loses 50% of its mass at 346.0 °C, while the pure blend loses the same amount of mass at 328.0 °C. Other selected antioxidants, including flavone and lignin, also improve the thermal stability of polylactide and polyhydroxybutyrate blend. In the case of PLA/P(3.4 HB) blend with flavone, the material loss of 50% takes place at the temperature of 335 °C, and in the case of the blend with lignin, this parameter amounted to 338.5 °C.

The thermal stability phenomenon can be seen more noticeably by using differential thermogravimetry (DTG) (Figure 3 and Table 4). According to DTG curves (Figure 3), it can also be observed that the thermal degradation of blends has two stages. The reason is that the first stage is related to P(3.4 HB) decomposing, and the second one is associated with the PLA degradation. The decomposing of P(3.4 HB) in a blend occurs at a higher temperature than in the case of pure material. This means that the presence of PLA in a blend delays the degradation of polyhydroxybutyrate by acting as a type of barrier, which was also verified by Arrieta et al. [37]. However, polylactide is more thermally stable than the P(3.4 HB) and PLA/P(3.4 HB) blend. Nevertheless, adding selected natural antioxidants, including flavone, *trans*-chalcone and lignin to the blend, can improve the thermal stability of the PLA/P(3.4 HB) material. According to the data obtained, the beginning of the pure blend’s main decomposing occurred at lower temperature compared to the blend with antioxidants. The most thermally stable blend contains *trans*-chalcone. This antioxidant changed blend’s beginning temperature of main decomposing (*T*_onset_) from 284.8 °C to 327.0 °C. The addition of other antioxidants, flavone and lignin caused an increase in this parameter to 311.4 °C and 318.7 °C respectively. Furthermore, the addition of flavone, *trans*-chalcone and lignin to PLA/P(3.4 HB) blend caused an increase in the temperature of the most intensive weight loss (*T*_d_) from 353.3 °C to 360.9 °C, 371.6 °C and 363.4 °C, respectively.

### 3.3. Dynamic Mechanical Analysis

Basing on the dynamic mechanical analysis results (Figure 4), it can be noticed that blending PLA with P(3.4 HB) at a proportion of 60/40 caused an interruption in PLA crystallization which is observed in the storage modulus decreasing. This phenomenon was confirmed also by Jung Seop Lim et al. in a polylactide blended with poly(3-hydroxybutyrate-co-3-hydroxyhexanoate) at different mass ratios [38]. Moreover, loss modulus of PLA also diminished significantly after mixing with P(3.4 HB) at the temperature of about 75 °C. Furthermore, pure PLA sample revealed an intensive peak of tanδ in a comparison to the blend. The reduced height of this peak in case of PLA/P(3.4 HB) blend was a result of the addition of polyhydroxybutyrate to polylactide, which caused restraint of the PLA chain mobility. The similar occurrence was proved also by Jung Seop Lim et al. [38]. Adding natural stabilizers to the blend did not cause changes in the dynamic mechanical analysis results as expected, because such stabilizers should not affect these properties.

### 3.4. Mechanical Tests

According to the mechanical test results illustrated in Figure 5, the blend of polylactide and polyhydroxybutyrate in proportion 60 and 40 is characterised by lower maximum stress results in comparison to reference material of PLA. This parameter decreased from 57.79 MPa to 37.12 MPa after adding polyhydroxybutyrate to polylactide. Also, strain at maximum stress (ε_M_) of PLA, which was 6.79%, was halved after adding polyhydroxybutyrate. As can be seen, the addition of P(3.4 HB) to polylactide reduced not only thermal parameters, but also mechanical ones. The reason is that too large an amount of P(3.4 HB) crystals could not be finely dispersed in PLA matrix in order to behave as a filler, as was proven by Arrieta et. al. [37] in the case of 75/25 PLA/PHB blend. Nevertheless, addition of natural antioxidants did not change σ_M_ and ε_M_ significantly.

For the tested samples, mechanical parameters were measured also after solar aging and the aging factor [30] was calculated based on the Equation (6):(6)$$A_{f}={\left(TS\cdot E_{B}\right)}_{after\;aging}/{\left(TS\cdot E_{B}\right)}_{after\;aging}$$
where *A_f_* is the aging coefficient, *TS* is tensile strength (MPa), and *E_B_* is the elongation at break (%).

According to Table 5, solar aging caused severe deterioration of tested materials mechanical properties. Based on the aging factor results, pure PLA proved to be the most stable in extreme conditions prevailing in solar simulator climate chamber. P(3.4 HB) and PLA/P(3.4 HB) are much more sensitive and even addition of natural antioxidants did not prevent materials from degradation. Selected antioxidants are good thermal stabilizers, but they are not sufficient in terms of other conditions, including higher humidity and solar radiation intensity.

### 3.5. Fourier Transform Infrared Spectroscopy (FT-IR) Absorbance Spectra Analysis

The –C–O (1187 cm^−1^) stretching band in PLA/P(3.4 HB) [31] is sensitive to solar aging process. which is observable in the Figure 6 During the solar aging. the absorbance of this band decreased in the case of pure P(3.4 HB) sample and its blend with polylactide and flavone. Only PLA/P(3.4 HB) with *trans*-chalcone blend turned out to be an exception and for this sample. where absorbance of the –C–O stretching band increased after solar aging. In the case of pure polylactide as well as PLA/P(3.4 HB) with lignin. changes in the absorbance did not occur significantly. A similar phenomenon is observed in the case of –C=O (1748 cm^−1^) and –CH_3_ (1451 cm^−1^) [31,32]. This means that only lignin reveals such stabilizing effect on surface PLA/P(3.4 HB) blend.

### 3.6. Colour Measurement

Some additives are able to change the colour of the polymer matrix, which can be sometimes considered as an advantage. Such natural stabilizers with colouring properties can be utilized as colorants for polymers. which was also proposed in literature [39]. On the other hand, in certain cases, the colour changing of polymer is not desirable, especially for packaging materials. where the transparency of the composite is infinitely preferable. Nevertheless, it is important to investigate the impact of selected stabilizers on material appearance. Colour measurement results presented in a bar chart (Figure 7) indicate that lignin changed the colour of the reference samples most significantly. The colour change index amounted to 34, which means that even an inexperienced observer is able to detect the difference between the test and reference samples. The PLA/P(3.4 HB) specimen containing lignin got the lowest whiteness index and was the darkest of all the materials. On the other hand, according to the data obtained, *trans*-chalcone also changed PLA/P(3.4 HB) blend, but in a different way. The addition of this antioxidant lightened the blend and lowered the chroma from about nine to three. This means that either lignin or *trans*-chalcone can be utilized as colorant. In contrast to these natural stabilizers, the third one, flavone, did not change the appearance of the blend. This compound seems to be the most appropriate additive with antioxidant and thermal stability properties dedicated for colourless packaging.

According to the results revealed in Figure 8, it can be observed that PLA/P(3.4 HB) blend with flavone and *trans*-chalcone changed their colour most significantly during solar aging. Their chroma (*C*_ab_) results got significantly higher and the whiteness index (*W*_i_) noticeably decreased. Depending on the point of view. This phenomenon can be treated as a drawaback or an asset. The colour changing of the tested sample during the aging process can be used as information about the degradation degree of the material and the flavone and *trans*-chalcone additives can be utilized as aging indicators, which is also noticeable in Figure 9. This idea has already been revealed in the case of β-carothene [6]. On the other hand, when the colour changing of a material during aging process is not needed. The choice of PLA/P(3.4 HB) blends with lignin would be the most appropriate because the colour of this composite is the most stable.

## 4. Conclusions

Nowadays, it is of considerable importance to create non-toxic, biobased, and biodegradable materials with environmentally friendly additives. This research focused on blends based on the main pro-ecological polymers, namely polylactide and poly(hydroxy butyrate) (P(3.4 HB)), with the addition of selected natural antioxidants, flavone, *trans*-chalcone, and lignin. According to the results obtained, blending polylactide (PLA) with polyhydroxybutyrate (P(3.4 HB)) caused a reduction of polylactide material’s mechanical and thermal properties. The summary of the revealed properties and application of the tested antioxidant are presented in Table 6. On the basis of the data obtained. selected antioxidants are able to improve the thermal properties of polymeric material by inhibiting its oxidation and sever decomposition at increased temperatures. This was confirmed by thermogravimetry and differential scanning calorimetry. Moreover, lignin as well as *trans*-chalcone are able to change the colour of blend significantly and may also be applied as colorants, in contrast to flavone which can be used to keep a material colorless. Therefore, selected natural antioxidants can be used as multifunctional and environmentally friendly additives for polymer blends of polylactide and polyhydroxybutyrate (P(3.4 HB)).

## Figures and Tables

**Figure 1 polymers-12-00074-f001:**
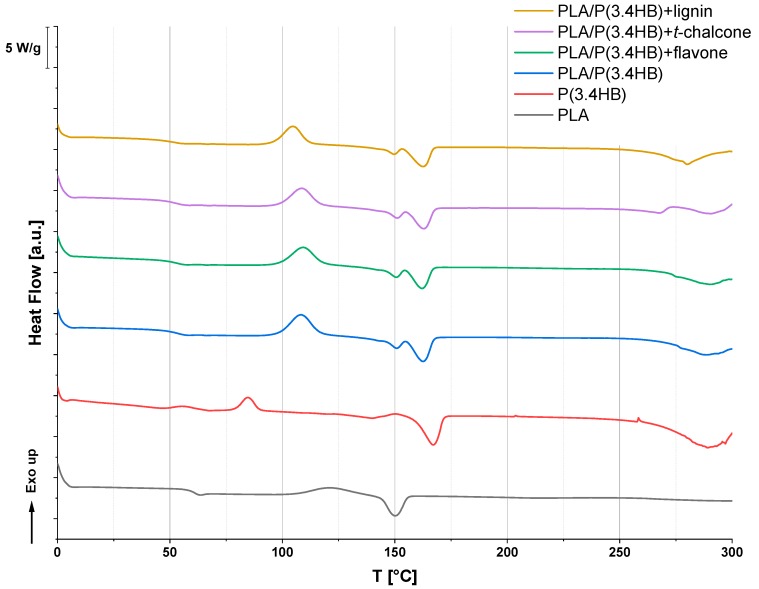
DSC curves of PLA/P(3.4 HB) composites with selected natural antioxidants in comparison to reference PLA, P(3.4 HB), and PLA/P(3.4 HB) samples.

**Figure 2 polymers-12-00074-f002:**
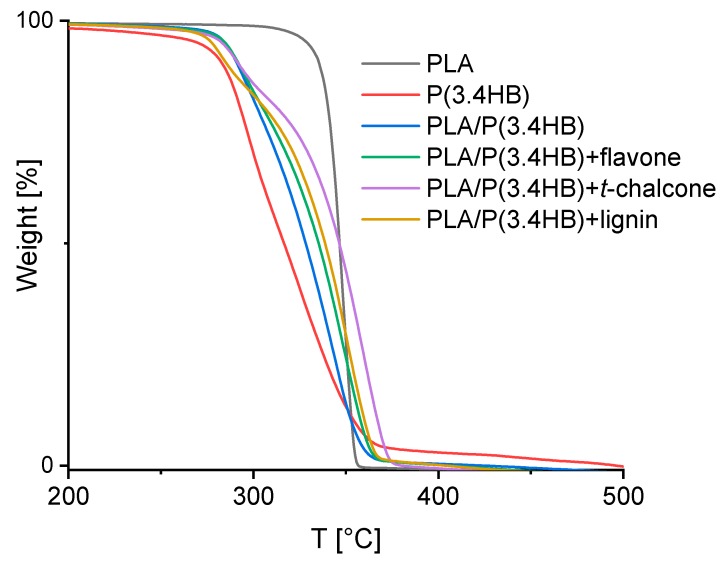
TGA curves of PLA/P(3.4 HB) composites with selected natural antioxidants in comparison to reference PLA, P(3.4 HB) and PLA/P(3.4 HB) samples.

**Figure 3 polymers-12-00074-f003:**
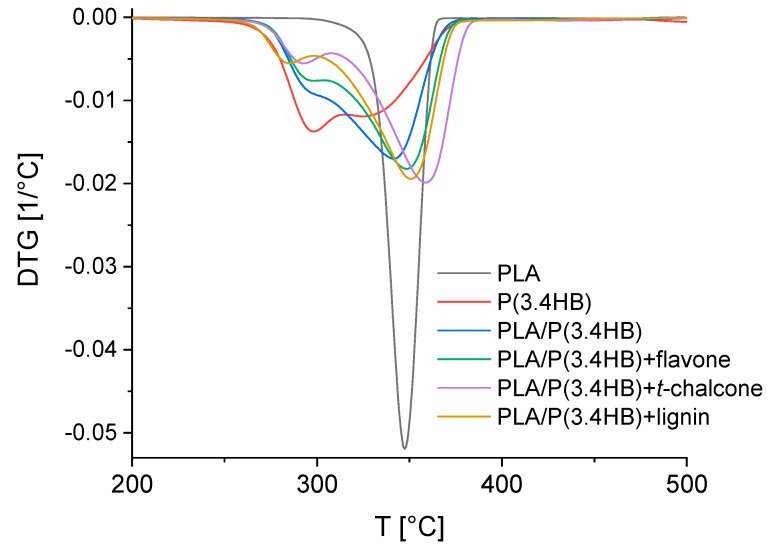
DTG curves of PLA/P(3.4 HB) composites with selected natural antioxidants in comparison to reference PLA, P(3.4 HB) and PLA/P(3.4 HB) samples.

**Figure 4 polymers-12-00074-f004:**
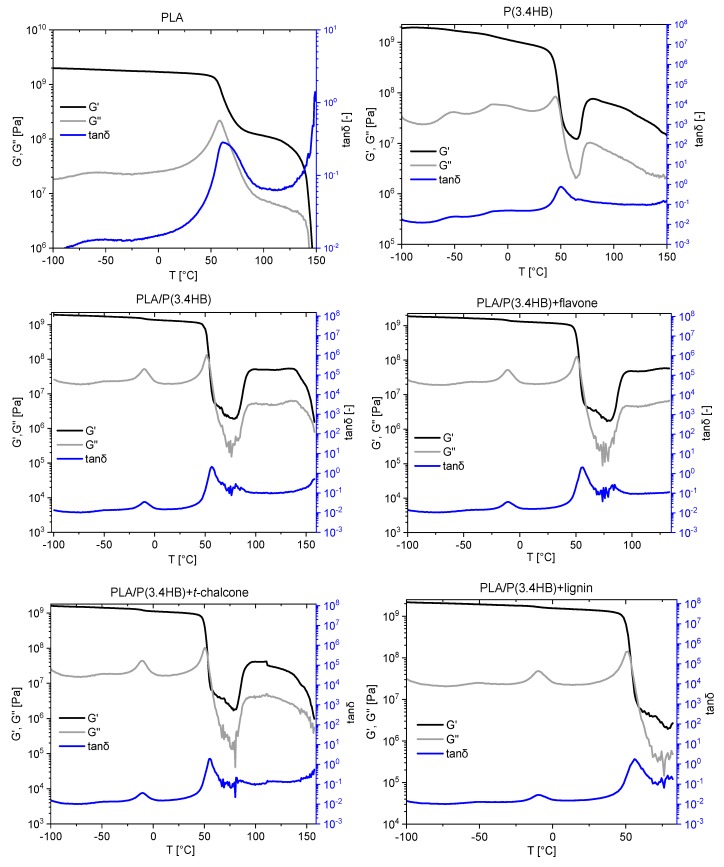
The temperature-dependent functions of storage modulus G′ (Pa), loss modulus G″ [Pa] and the loss factor tanδ for PLA, P(3.4 HB) as well as blends with natural antioxidants.

**Figure 5 polymers-12-00074-f005:**
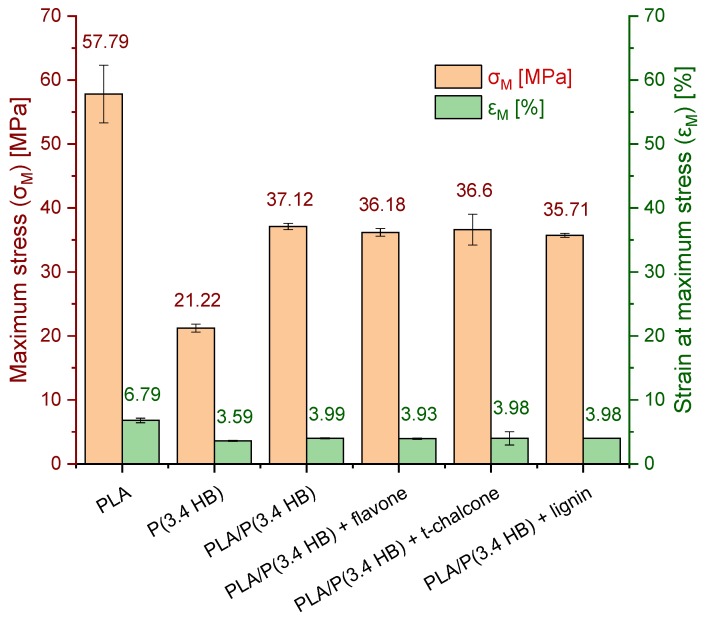
Maximum stress (σ_M_) (MPa) and strain at maximum stress (ε_M_) (%) results of PLA/P(3.4 HB) composites with selected natural antioxidants in comparison to reference PLA, P(3.4 HB) and PLA/P(3.4 HB) samples.

**Figure 6 polymers-12-00074-f006:**
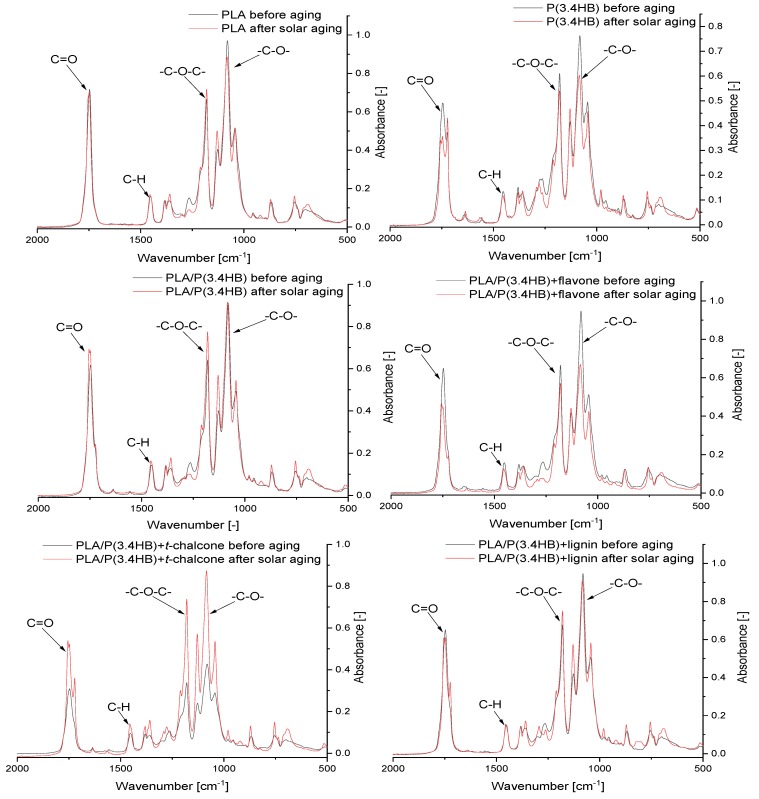
FT-IR spectra of PLA, P(3.4 HB), and blends with antioxidants before and after solar aging.

**Figure 7 polymers-12-00074-f007:**
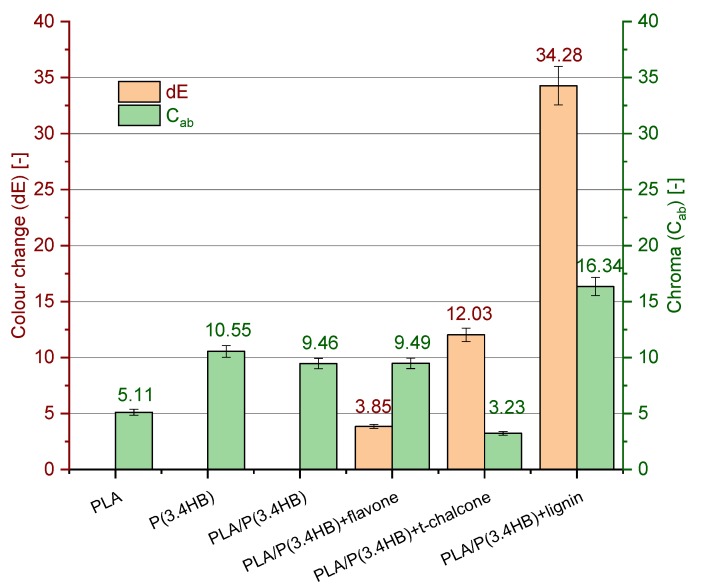
Colour change (dE) (-) and Chroma (*C*_ab_) (-) results of PLA/P(3.4 HB) composites with selected natural antioxidants in comparison to reference PLA. P(3.4 HB) and PLA/P(3.4 HB) samples.

**Figure 8 polymers-12-00074-f008:**
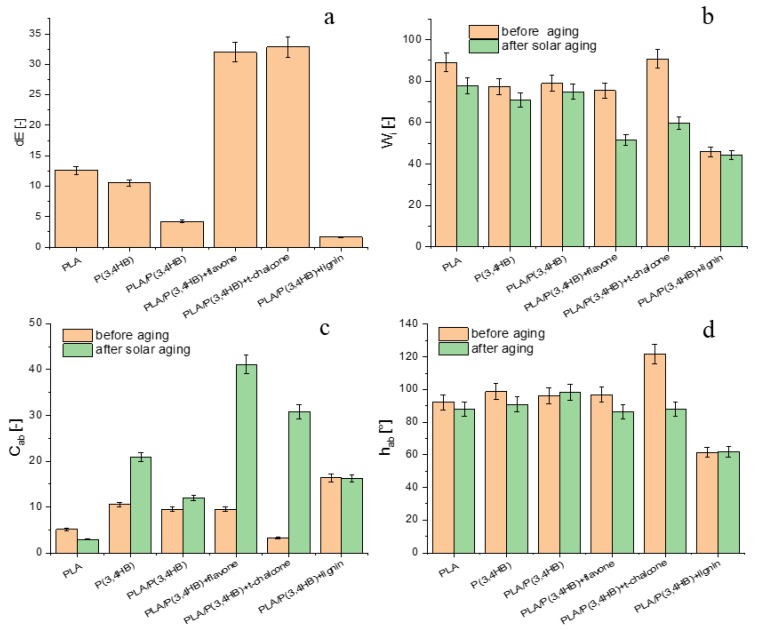
The impact of solar aging on: (**a**) change of colour (dE). (**b**) whiteness index (*W*_i_). (**c**) chroma (*C*_ab_). (**d**) hue angle (*h*_ab_) of aliphatic polyester composites containing natural antioxidants.

**Figure 9 polymers-12-00074-f009:**
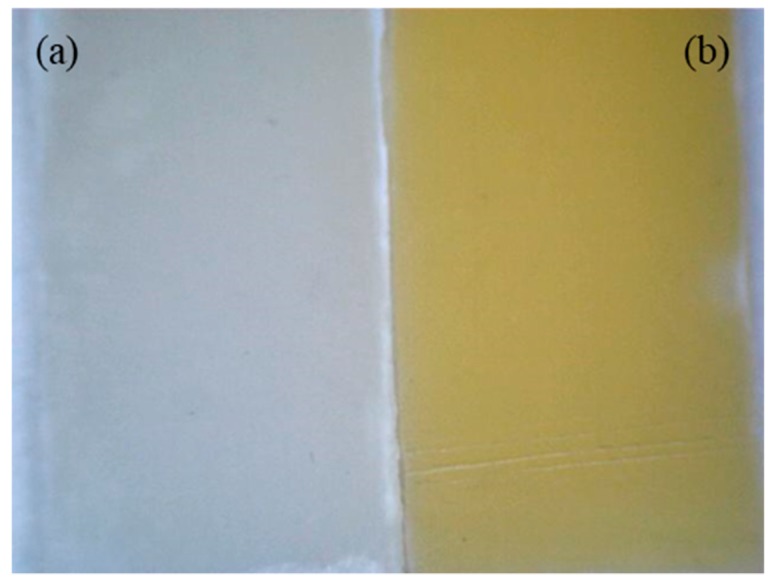
Picture of PLA/P(3.4 HB) with flavone sample before (**a**) and after (**b**) solar aging.

**Table 1 polymers-12-00074-t001:** The composition of prepared polylactide/polyhydroxybutyrate samples containing natural antioxidants.

Components	Weight Composition [phr]					
	PLA	P(3.4 HB)	PLA/P(3.4 HB)	PLA/P(3.4 HB) + Flavone	PLA/P(3.4 HB) + *t*-Chalcone	PLA/P(3.4 HB) + Lignin
Polylactic acid (PLA)	100	-	60	60	60	60
Polyhydroxybutyrate (P(3.4 HB))	-	100	40	40	40	40
flavone	-	-	-	1.25	-	-
*trans*-chalcone	-	-	-	-	1.25	-
lignin	-	-	-	-	-	1.25

phr—part per hundred resin.

**Table 2 polymers-12-00074-t002:** Differential scanning calorimetry (DSC) analysis of PLA/PHB samples contained selected antioxidants.

Sample	*Tg* (°C)	Δ*Hcc* (°C)	*Tcc* (°C)	Δ*Hm* (J/g)	*Tm* (°C)	Δ*Ho* (J/g)	*To* (°C)
PLA	60.9	18.0	121.4	14.8	150.1	4.9	226.0
P(3.4 HB)	47.4	12.3	84.6	6.8	139.8	0.7	205.7
				31.6	166.7		
PLA/P(3.4 HB)	54.8	24.4	108.4	30.0	162.2	5.5	232.7
PLA/P(3.4 HB) + flavone	53.5	24.6	109.2	29.1	161.9	3.3	241.1
PLA/P(3.4 HB) + *t*-chalcone	54.2	23.5	108.5	28.8	162.6	8.0	268.6
PLA/P(3.4 HB) + lignin	52.1	24.0	104.7	30.3	162.3	5.0	224.8

**Table 3 polymers-12-00074-t003:** Temperatures of the tested materials weight loss. *T*_x%_ is temperature at which the weight loss is x%.

Sample	*T*_2%_ (°C)	*T*_5%_ (°C)	*T*_10%_ (°C)	*T*_15%_ (°C)	*T*_20%_ (°C)	*T*_50%_ (°C)	*T*_70%_ (°C)	*T*_90%_ (°C)	*T*_100%_ (°C)
PLA	314.5	328.7	335.5	338.5	340.7	346.7	349.8	352.7	357.3
P(3.4 HB)	214.8	270.3	283.8	289.0	292.7	316.8	333.3	354.3	496.8
PLA/P(3.4 HB)	270.3	284.5	291.3	297.3	302.5	328.0	340.8	352.8	424.8
PLA/P(3.4 HB) + flavone	265.0	284.5	292.0	298.8	305.5	335.5	346.8	358.0	404.5
PLA/P(3.4 HB) + *t*-chalcone	256.0	283.0	292.8	301.8	313.8	346.0	357.3	367.0	380.5
PLA/P(3.4 HB) + lignin	259.8	277.8	286.8	296.5	307.0	338.5	349.8	360.3	403.8

**Table 4 polymers-12-00074-t004:** DTG results for pure PLA, P(3.4 HB), PLA/P(3.4 HB) blend and PLA/P(3.4 HB) blends with selected natural antioxidants (flavone, *trans*-chalcone and lignin): temperature of beginning of main decomposing (*T*_onset_), temperature of the most intensive weight loss (*T*_d_).

Sample	*T*_onset_ (°C)	*T*_d_ (°C)
PLA	352.7	357.3
P(3.4 HB)	283.3	308.6
PLA/P(3.4 HB)	284.8	353.3
PLA/P(3.4 HB) + flavone	311.4	360.9
PLA/P(3.4 HB) + *t*-chalcone	327.0	371.6
PLA/P(3.4 HB) + lignin	318.7	363.4

**Table 5 polymers-12-00074-t005:** The comparison of mechanical properties of tested PLA, P(3.4 HB), and PLA/P(3.4 HB) composites before and after solar aging.

Sample	Before Solar Aging		After Solar Aging		A*_f_* (−)
	σ_B_ (MPa)	ε_B_ (%)	σ_B_ (MPa)	ε_B_ (%)	
PLA	53.24	8.28	13.76	2.77	0.086
P(3.4 HB)	14.21	10.53	1.74	0.47	0.005
PLA/P(3.4 HB)	19.71	61.57	2.23	2.59	0.005
PLA/P(3.4 HB) + flavone	21.79	47.02	2.95	0.99	0.003
PLA/P(3.4 HB) + *t*-chalcone	23.28	78.00	4.25	2.95	0.007
PLA/P(3.4 HB) + lignin	22.88	38.90	2.97	1.21	0.004

**Table 6 polymers-12-00074-t006:** Summary of revealed properties and application of tested antioxidant.

**flavone**	does not change colour of polymer	can be applied as aging indicator	can be used as thermal stabilizer
***trans*-chalcone**	can be utilized as colorant for polymer	can be applied as aging indicator	can be used as thermal stabilizer
**lignin**	can be utilized as colorant for polymer	does not change colour of polymer during aging process	can be used as thermal stabilizer

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
