# Peer review of "Thermal Analysis of Aliphatic Polyester Blends with Natural Antioxidants"

_polymers, 2020, doi:10.3390/polym12010074_

Round 1

Reviewer 1 Report

Manuscript entitled “Thermal analysis of aliphatic polyester blends with natural antioxidants”

In this paper the authors study the application of antioxidants of plant origin (flavone, trans-chalcone and lignin) to protect polylactide/polyhydroxybutyrate blends, inin order to obtain fully biobased  and biodegradable materials with application in packaging. DSC results indicate that these natural antioxidants increase the temperature of oxidation of the blend. Also, by TGA they also obtained an increase in thermal stability.  In addition, they explore the possibility of using these antioxidants as colorants or aging indicators,

I find that the subject of the paper is of interest, however I found that in the section of results there some aspects that should be improved. Therefore, I think the paper could be published but after major revision addressing some points that I explain in the following:

Major amendments:

1- Table 1: In my opinion, M1 to M6 codes should be removed from the table because they are not used in any other part of the manuscript.

2- The aging coefficient is named in different ways in several parts of the manuscript. In lines 165, 167 and 288 it is named as” K”. However, in lines 168, 289 and Table 3 “Af” is used to name aging coefficient.   A consistent nomenclature should be used.

3- Figure 1: It is very difficult to distinguish the DSC curves of different samples. I would suggest to shift vertically the curves, keeping a distance between them in order to clarify the interpretation.

4- Table 2: The columns containing melting enthalpy and melting temperature values are confusing and it is not very clear, at least for me, to which sample correspond some values (for example, values of melting enthalpy 6.8 and 31.6).

5- Section 3.2: the analysis of TGA results should be improved. It has to be indicated which is the temperature taken as the initiation of degradation (Tonset, T2%, T5%...). And it would be easier to follow the explanation of TGA results if a table was added. This table could include the values of the temperature chosen as initiation of degradation, and also other values of TGA as temperature of maximum degradation, residue at 500 ºC...

6- In line 262: it is not clear for me the sentence “It means, that addition of poly(3-hydroxybutyrate-co-3-hydroxyhexanoate) to polylactide caused inhibition of the PLA chain mobility”.

7- In line 355: authors conclude that trans-chalcone acted like a plasticizer and caused increase in elongation at break. This result has not been explained in section 3.4. In addition, in section 3.1 the results of DSC analysis show that only lignin has a plasticizing effect.

Minor amendments

8- Line 17: According to International Association of Thermal Analysis, “thermogram” word can only be used for TGA curves. For DSC, “DSC runs” or ·”DSC curves” should be used. The use of the “thermogram” word applied to DSC curves, should be corrected in several parts of the manuscript

9- When indicating temperatures a space should be left between the number and the unit: example: line 107 instead of “55-57ºC” it should be written ““55-57 ºC”. This has to be corrected all over the manuscript.

10- Section 2.7: the number of individual samples used to determine the average values should be indicated.

11- Section 2.8: resolution and number of scans for FTIR measurements should be given.

12- Figure 4: the title of each of the figures has a size that is too big, I think that using a smaller font size would increase the visual quality.

13- Figure 5: Again, the size used for the values of maximum stress and strain should be reduced.

14- Line 274: it is written “Acording to the mechanical test results illustrated in Figure 4.” And it should be written “ Acording to the mechanical test results illustrated in Figure 5.”

15- Table 3: Values should be represented with decimal points instead of comma.

16- In Figure 6: much smaller font size should be used for the assignement of bands.

Author Response

Institute of Polymer and Dye Technology

Technical University of Lodz

90-924 Lodz, ul Stefanowskiego 12/16, Poland

Tel.: +48 42 631 32 23, Fax: +48 42 636 25 43

December 13, 2019

Polymers

Dear Professor,

We are resubmitting our revised paper entitled Thermal analysis of aliphatic polyester blends with natural antioxidants by, Olga Olejnik, Anna Masek with a request to reconsider it for publication in Polymers.

We have carefully considered the Editor and Reviewers' comments. The manuscript was revised exactly according to these comments. The list of responses to the reviewer’s comments and corrections made in the manuscript is attached.

The manuscript has not been previously published, is not currently submitted for review to any other journal, and will not be submitted elsewhere before a decision is made by this journal.

For correspondence please use the following information:

corresponding author: Anna Masek

Institute of Polymer and Dye Technology

Technical University of Lodz

90-924 Lodz, ul Stefanowskiego 12/16, Poland

Tel.: +48 42 631 32 93

Fax: +48 42 636 25 43

Yours sincerely,

Ph. D., D.Sc. Anna Masek

Answers to reviewer #1 comments

Reviewer #1: Manuscript entitled “Thermal analysis of aliphatic polyester blends with natural antioxidants” In this paper the authors study the application of antioxidants of plant origin (flavone, trans-chalcone and lignin) to protect polylactide/polyhydroxybutyrate blends, inin order to obtain fully biobased and biodegradable materials with application in packaging. DSC results indicate that these natural antioxidants increase the temperature of oxidation of the blend. Also, by TGA they also obtained an increase in thermal stability. In addition, they explore the possibility of using these antioxidants as colorants or aging indicators, I find that the subject of the paper is of interest, however I found that in the section of results there some aspects that should be improved. Therefore, I think the paper could be published but after major revision addressing some points that I explain in the following:

Major amendments:

Table 1: In my opinion, M1 to M6 codes should be removed from the table because they are not used in any other part of the manuscript.

Answer: We are thankful for Reviewer’s comment. M1 to M6 codes have been replaced by names of samples used in the text (PLA, P(3,4HB), PLA/P(3,4HB), PLA/P(3,4HB)+flavone, PLA/P(3,4HB)+t-chalcone, PLA/P(3,4HB)+lignin). Table 1. has been also modified according to the comment of Reviewer #2.

Components:

Weight composition [phr]

PLA

P(3,4HB)

PLA/P(3,4HB)

PLA/P(3,4HB)

+flavone

PLA/P(3,4HB)

+t-chalcone

PLA/P(3,4HB)

+lignin

Polylactic acid (PLA)

100

-

60

60

60

60

Polyhydroxybutyrate (P(3,4 HB))

-

100

40

40

40

40

flavone

-

-

-

1.25

-

-

trans-chalcone

-

-

-

-

1.25

-

lignin

-

-

-

-

-

1.25

The aging coefficient is named in different ways in several parts of the manuscript. In lines 165, 167 and 288 it is named as” K”. However, in lines 168, 289 and Table 3 “Af” is used to name aging coefficient. A consistent nomenclature should be used.

Answer: This mistake has been corrected and now in every part of text “Af” indication for aging coefficient is used.

Figure 1: It is very difficult to distinguish the DSC curves of different samples. I would suggest to shift vertically the curves, keeping a distance between them in order to clarify the interpretation.

Answer: We are thankful for this comment. This Figure has been changed into more clear with curves vertically shifted keeping a distance between them.

Table 2: The columns containing melting enthalpy and melting temperature values are confusing and it is not very clear, at least for me, to which sample correspond some values (for example, values of melting enthalpy 6.8 and 31.6).

Answer: We are grateful for Reviewer’s comment. This Table has been modified into more clear one by adding extra lines and grey colour.

Sample

Tg [°C]

ΔHcc [°C]

Tcc [°C]

ΔHm [J/g]

Tm [°C]

ΔHo [J/g]

To [°C]

PLA

60.9

18.0

121.4

14.8

150.1

4.9

226.0

P(3,4HB)

47.4

12.3

84.6

6.8

139.8

0.7

205.7

31.6

166.7

PLA/P(3,4HB)

54.8

24.4

108.4

30.0

162.2

5.5

232.7

PLA/P(3,4HB)+flavone

53.5

24.6

109.2

29.1

161.9

3.3

241.1

PLA/P(3,4HB) +

t-chalcone

54.2

23.5

108.5

28.8

162.6

8.0

268.6

PLA/P(3,4HB) + lignin

52.1

24.0

104.7

30.3

162.3

5.0

224.8

Section 3.2: the analysis of TGA results should be improved. It has to be indicated which is the temperature taken as the initiation of degradation (Tonset, T2%, T5%...). And it would be easier to follow the explanation of TGA results if a table was added. This table could include the values of the temperature chosen as initiation of degradation, and also other values of TGA as temperature of maximum degradation, residue at 500 ºC...

Answer: We thank Reviewer for paying attention to this problem. We have tried to improve presentation of results and make it more clearly. We have added two extra Tables and described this paragraph once again as follow:

Thermogravimetric Analysis (TGA)

Thermogravimetric analysis allowed a determination of material thermal stability. According to the TGA thermogram (Figure 2.) and Table 3, blending PLA with P(3,4HB) reduces the resistance of PLA to high temperatures. The blend lost 5% of its weight at a temperature of 284.5 °C, while the temperature of 5% PLA decomposition amounted to 328.7 °C. Adding polyphenols to PLA/P(3,4HB) blend caused the improvement in thermal stability At first glance, the stability is not noticeable and PLA/P(3,4HB) blend with antioxidants loses 5% of its weight at a similar temperature as the pure blend. Nevertheless, the added natural stabilizers subsequently slowed down the blend decomposition. This phenomenon starts to occur in case of 15% material weight loss and is the most visible for 50% of sample weight loss. According to the gathered data, PLA/P(3,4HB) with trans-chalcone is the most thermally stable and loses 50% of its mass at 346.0 °C, while the pure blend loses the same amount of mass at 328.0 °C. Other selected antioxidants, including flavone and lignin, also improve the thermal stability of polylactide and polyhydroxybutyrate blend. In the case of PLA/P(3,4HB) blend with flavone, the material loss of 50% takes place at the temperature of 335 °C, and in the case of the blend with lignin, this parameter amounted to 338.5 °C.

Figure 2. TGA curves of PLA/P(3,4HB) composites with selected natural antioxidants in comparison to reference PLA, P(3,4HB) and PLA/P(3,4HB) samples.

Table 3. Temperatures of the tested materials weight loss. Tx% - temperature at which the weight loss is x%.

Sample

T2% [°C]

T5% [°C]

T10% [°C]

T15% [°C]

T20% [°C]

T50% [°C]

T70% [°C]

T90% [°C]

T100% [°C]

PLA

314.5

328.7

335.5

338.5

340.7

346.7

349.8

352.7

357.3

P(3,4HB)

214.8

270.3

283.8

289.0

292.7

316.8

333.3

354.3

496.8

PLA/P(3,4HB)

270.3

284.5

291.3

297.3

302.5

328.0

340.8

352.8

424.8

PLA/P(3,4HB)
+flavone

265.0

284.5

292.0

298.8

305.5

335.5

346.8

358.0

404.5

PLA/P(3,4HB)
+t-chalcone

256.0

283.0

292.8

301.8

313.8

346.0

357.3

367.0

380.5

PLA/P(3,4HB)
+lignin

259.8

277.8

286.8

296.5

307.0

338.5

349.8

360.3

403.8

The thermal stability phenomenon can be seen more noticeably by using differential thermogravimetry (DTG) (Figure 3. and Table 4.). According to DTG curves (Figure 3.), it can also be observed that the thermal degradation of blends has two stages. The reason is that the first stage is related to P(3,4HB) decomposing, and the second one is associated with the PLA degradation. The decomposing of P(3,4HB) in a blend occurs at a higher temperature than in the case of pure material. This means that the presence of PLA in a blend delays the degradation of polyhydroxybutyrate by acting as a type of barrier, which was also verified by Arrieta et. al. [39]. However, polylactide is more thermally stable than P(3,4HB) and PLA/P(3,4HB) blend. Nevertheless, adding selected natural antioxidants, including flavone, trans-chalcone and lignin to the blend, can improve the thermal stability of the PLA/P(3,4HB) material. According to the data obtained, the beginning of the pure blend’s main decomposing occurred at lower temperature compared to the blend with antioxidants. The most thermally stable blend contains trans-chalcone. This antioxidant changed blend’s beginning temperature of main decomposing (Tonset) from 284.8 °C to 327.0 °C. Addition of other antioxidants, flavone and lignin caused an increase in this parameter to 311.4 °C and 318.7° C respectively. Furthermore, the addition of flavone, trans-chalcone and lignin to PLA/P(3,4HB) blend caused an increase in the temperature of the most intensive weight loss (Td) from 353.3 °C to 360.9 °C, 371.6 °C and 363.4 °C respectively.

Table 4. DTG results for pure PLA, P(3,4HB), PLA/P(3,4HB) blend and PLA/P(3,4HB) blends with selected natural antioxidants (flavone, trans-chalcone and lignin): temperature of beginning of main decomposing (Tonset), temperature of the most intensive weight loss (Td)

Sample

Tonset [°C]

Td [°C]

PLA

352.7

357.3

P(3,4HB)

283.3

308.6

PLA/P(3,4HB)

284.8

353.3

PLA/P(3,4HB)
+flavone

311.4

360.9

PLA/P(3,4HB)
+t-chalcone

327.0

371.6

PLA/P(3,4HB)
+lignin

318.7

363.4

Figure 3. DTG curves of PLA/P(3,4HB) composites with selected natural antioxidants in comparison to reference PLA, P(3,4HB) and PLA/P(3,4HB) samples.

In line 262: it is not clear for me the sentence “It means, that addition of poly(3-hydroxybutyrate-co-3-hydroxyhexanoate) to polylactide caused inhibition of the PLA chain mobility”.

Answer: presented sentence was a continuation of following part: “Furthermore, pure PLA sample revealed an intensive peak of tanδ in a comparison to the blend.” and we have tried to explain this part by cited sentence. We have tried to replace unclear presented sentence into following: “The reduced height of this peak in case of PLA/P(3,4HB) blend was a result of the addition of polyhydroxybutyrate to polylactide, which caused restraint of the PLA chain mobility.”

In line 355: authors conclude that trans-chalcone acted like a plasticizer and caused increase in elongation at break. This result has not been explained in section 3.4. In addition, in section 3.1 the results of DSC analysis show that only lignin has a plasticizing effect.

Answer: We thank Reviewer for paying attention to this issue. We realized that the plasticizing effect of presented substances is not strongly proved, thus we decided to remove it as conclusion. This effect should be confirmed by more than technique.

Minor amendments

Line 17: According to International Association of Thermal Analysis, “thermogram” word can only be used for TGA curves. For DSC, “DSC runs” or ·”DSC curves” should be used. The use of the “thermogram” word applied to DSC curves, should be corrected in several parts of the manuscript

Answer: All “thermogram” words in case of DSC analysis have been replaced by “DSC curves”

When indicating temperatures a space should be left between the number and the unit: example: line 107 instead of “55-57ºC” it should be written ““55-57 ºC”. This has to be corrected all over the manuscript.

Answer: The mistake has been corrected.

Section 2.7: the number of individual samples used to determine the average values should be indicated.

Answer: The missing information has been placed in section 2.7. “The number of individual samples used in this research was 6 for every tested material”

Section 2.8: resolution and number of scans for FTIR measurements should be given.

Answer: The missing information has been placed in section 2.8. “The number of used scans amounted to 64

Figure 4: the title of each of the figures has a size that is too big, I think that using a smaller font size would increase the visual quality.

Answer: The size of every title in Figure 4. has been reduced.

Figure 5: Again, the size used for the values of maximum stress and strain should be reduced.

Line 274: it is written “Acording to the mechanical test results illustrated in Figure 4.” And it should be written “ Acording to the mechanical test results illustrated in Figure 5.”

Answer: The mistake has been corrected.

Table 3: Values should be represented with decimal points instead of comma.

Answer: The mistake has been corrected.

In Figure 6: much smaller font size should be used for the assignement of bands.

Answer: The font size of assignment bands has been reduced.

Reviewer 2 Report

This manuscript deals with aliphatic polyester blends with natural antioxidants and the samples were characterized with DSC, TGA and dynamical mechanical analysis. Mechanical tests were also performed, and also FTIR analysis and color measurement. The paper needs some editing of English language and mainly style revisions are required. The scientific findings are interesting.
I suggest a few revisions that should improve the quality of the manuscript:

The degree symbol should have always a space after the number, as in 200 ºC. Cistus Linnaeus and juglans regia Linnaeus should appear in italics. There is always a space between the numerical value and unit symbol, as in 24 J/g. In line 45, there is a double space between “phytochemicals” and “also”. Line 99 – (P(3,4HB) -> (P(3,4HB)) Line 110 -> The SI symbol for minute or minutes is min (without a dot). The same with rpm. Table 1- The legend will be more clearly if it is: “Weight composition (%) of the prepared samples (M1, M2, M3, M4, M5, M6) containing polylactide (PLA), polyhydroxybutyrate (P(3,4 HB)) and natural antioxidants (flavone, trans-chalcone and lignin)). “Weight ratio” doesn’t make sense in this case. It’s not a ratio, it’s a percentage of composition. Line 167: Where is the aging coefficient (Af) in the equation? I think it will add value to the paper if you add a summary table with ALL the advantages/disadvantages of the different antioxidants, because the conclusions are very incomplete since you used a lot of techniques.

Flavone

Keep transparency of the materials

Trans-chalcone

Can be applied as colorant

Acted like a plasticizer

Lignin

Can be applied as colorant

(…)

 Line 362: “This research was funded”

Author Response

Institute of Polymer and Dye Technology

Technical University of Lodz

90-924 Lodz, ul Stefanowskiego 12/16, Poland

Tel.: +48 42 631 32 23, Fax: +48 42 636 25 43

December 13, 2019

Polymers

Dear Professor,

We are resubmitting our revised paper entitled Thermal analysis of aliphatic polyester blends with natural antioxidants by, Olga Olejnik, Anna Masek with a request to reconsider it for publication in Polymers.

We have carefully considered the Editor and Reviewers' comments. The manuscript was revised exactly according to these comments. The list of responses to the reviewer’s comments and corrections made in the manuscript is attached.

The manuscript has not been previously published, is not currently submitted for review to any other journal, and will not be submitted elsewhere before a decision is made by this journal.

For correspondence please use the following information:

corresponding author: Anna Masek

Institute of Polymer and Dye Technology

Technical University of Lodz

90-924 Lodz, ul Stefanowskiego 12/16, Poland

Tel.: +48 42 631 32 93

Fax: +48 42 636 25 43

Yours sincerely,

Ph. D., D.Sc. Anna Masek

Answers to reviewer #2 comments

Reviewer #2: This is a carefully done study which merits publication. In order
to emphasize the significance of this work to readers, the authors might want to add
the meaning of the obtained data and make adequate revisions. The following is a list
of my comments, including those points and other noticed ones.

The comments are listed below.

This manuscript deals with aliphatic polyester blends with natural antioxidants and the samples were characterized with DSC, TGA and dynamical mechanical analysis. Mechanical tests were also performed, and also FTIR analysis and color measurement. The paper needs some editing of English language and mainly style revisions are required. The scientific findings are interesting.

I suggest a few revisions that should improve the quality of the manuscript:

The degree symbol should have always a space after the number, as in 200 ºC. Cistus Linnaeus and juglans regia Linnaeus should appear in italics. There is always a space between the numerical value and unit symbol, as in 24 J/g. In line 45, there is a double space between “phytochemicals” and “also”. Line 99 – (P(3,4HB) -> (P(3,4HB)) Line 110 -> The SI symbol for minute or minutes is min (without a dot). The same with rpm.

Answer: All mistakes have been corrected. Only rpm. has not be changed, because the dot we placed in the end of the sentence.

Table 1- The legend will be more clearly if it is: “Weight composition (%) of the prepared samples (M1, M2, M3, M4, M5, M6) containing polylactide (PLA), polyhydroxybutyrate (P(3,4 HB)) and natural antioxidants (flavone, trans-chalcone and lignin)). “Weight ratio” doesn’t make sense in this case. It’s not a ratio, it’s a percentage of composition.

Answer: Table 1 has been improved. Weight composition (%) can not be used, because it is not a percentage of composition, is a proportion. I have changed unit into [phr] which means “per hundred resin”.

Line 167: Where is the aging coefficient (Af) in the equation? I think it will add value to the paper if you add a summary table with ALL the advantages/disadvantages of the different antioxidants, because the conclusions are very incomplete since you used a lot of techniques.

Flavone

Keep transparency of the materials

Trans-chalcone

Can be applied as colorant

Acted like a plasticizer

Lignin

Can be applied as colorant

(…)

Answer: We are thankful for Reviewer’s advice. Table has been prepared as below:. Nevertheless it is hard to definitely determine advantages/disadvantages, because some of disadvantages can be treated as advantages in some cases. We decided to remove “plasticizing effect” of different antioxidants, because this effect should be confirmed by more than technique.

flavone

Does not change colour of polymer

Can be applied as aging indicator

Can be used as thermal stabilizer

trans-chalcone

Can be utilized as colorant for polymer

Can be applied as aging indicator

Can be used as thermal stabilizer

lignin

Can be utilized as colorant for polymer

Does not change colour of polymer during aging process

Can be used as thermal stabilizer

 Line 362: “This research was funded”

Answer: The mistake has been corrected.

Round 2

Reviewer 1 Report

I consider that my recomendations have been correctly adressed in the second version of the manuscript and, therefore, I think that this new version can be published as it is.

Note: there is a problem int Table 1, some numbers appear on top of the table. But, it may be a problem of pdf version.

This manuscript is a resubmission of an earlier submission. The following is a list of the peer review reports and author responses from that submission.